# Clinical outcomes of gastrointestinal bleeding management during anticoagulation therapy

Ho-Jun Jang[1], Dongyoung Lee[2], Tae-Hoon Kim[3]*, Je Sang Kim[4], Hyun-Jong Lee[1], Ji Bak Kim[5], Ji-young Kim[6]

**1** Division of Cardiology, Sejong General Hospital, Bucheon, Republic of Korea, **2** Division of Cardiology, Chamjoeun Hospital, Gwangju-si, Republic of Korea, **3** Division of Cardiology, CHA Ilsan Medical Center, CHA University School of Medicine, Goyang, Republic of Korea, **4** Division of Cardiology, Cardiovascular Center, Dongguk University Ilsan Hospital, Goyang, Republic of Korea, **5** Division of Cardiology, Korea University Guro Hospital, Seoul, Republic of Korea, **6** Department of Neurology, Inje University Seoul Paik Hospital, Seoul, Republic of Korea

☺ These authors contributed equally to this work.
* sch.kimtaehoon@gmail.com

**Data Availability Statement:** All relevant data are within the paper and its Supporting Information files.

## Abstract

### Background

Acute gastrointestinal (GI) bleeding is not an uncommon complication of oral anticoagulation (OAC) therapy that requires medication cessation. However, drug cessation may cause fatal stroke or systemic embolization in patients at high thromboembolic risk. Here we sought to find an appropriate anticoagulation cessation strategy in cases of GI bleeding during OAC therapy.

### Methods

This single-center retrospective cohort analysis was performed between 2010 and 2018. Patients were enrolled if the following three consecutive conditions were met: 1) electrocardiography electrocardiography-proven atrial fibrillation; 2) OAC therapy; and 3) GI bleeding. We divided the drug cessation strategy into the continuation and discontinuation groups. During 1-year follow-up, the rates of major thromboembolic and rebleeding events were calculated.

### Results

One hundred and forty-six patients (continuation [n = 54] vs. discontinuation [n = 92] group) were enrolled. Patients in the discontinuation group were more likely to be older (69.8 ± 9.0 yrs vs. 74.9 ± 8.9 yrs, p = 0.001), while patients in the continuation group were more likely to have undergone cardiac valve surgery (51.9% vs. 20.7%, p<0.001). The presence of a mechanical mitral valve was a determinant of continuation strategy (38.9% vs. 7.5%, p<0.001). However, the mean $CHA_2DS_2$-VASc (3.4±1.3 vs. 4.1±1.6, p = 0.010) and Glasgow-Blatchford (8.0±2.4 vs. 8.9±2.5, p = 0.037) scores were higher in the discontinuation group.

Two major embolic strokes occurred in each group (3.7% vs. 2.2%, p = 0.585). Four of 54 (7.4%) and five of 92 (5.4%) patients had rebleeding events during follow-up (p = 0.632).

**Funding:** The authors received no specific funding for this work.

**Competing interests:** The authors have declared that no competing interests exist.

One embolic event in the continuation group and one rebleeding event in the discontinuation group were fatal. The Glasgow-Blatchford score was a predictor of 1-year rebleeding events (odds ratio [OR], 1.36; 95% confidence interval [CI], 0.68–2.20; p = 0.028). The high $CHA_2DS_2$-VASc score showed a strong trend (OR, 1.71; 95% CI, 0.92–3.20; p = 0.089) in 1-year thromboembolic events.

## Conclusion

No single risk factor or drug cessation strategy was attributed to adverse clinical events after GI bleeding. The risk of future thrombotic or rebleeding events should be individualized and controlled for based on a pre-existing stratification system.

## Introduction

Gastrointestinal (GI) bleeding is not an uncommon complication in patients receiving anticoagulant therapy. If the thromboembolic risk is low enough, hemostasis may be achieved with the mere interruption of anticoagulation therapy. However, in patients at high thromboembolic risk, defined as those with a high $CHA_2DS_2$-VASc score, or with mechanical cardiac valves, the cessation of anticoagulation may cause fatal stroke or systemic embolization. It is necessary to measure the prothrombin time (INR) for patients taking warfarin or the duration of the reversal period of anticoagulant in the balance [1]. However, the choice of hemostasis strategy may be more complex because patients with a high thromboembolic risk tend to bleed more. Popular scoring system using $CHA_2DS_2$-VASc and HAS-BLED scores for thrombosis and bleeding shares many components [2,3]. Thus, this study aimed to evaluate patient prognosis according to anticoagulation cessation strategy when GI bleeding occurred during oral anticoagulation (OAC) therapy.

## Methods

### Data acquisition

We retrospectively analyzed patients who visited Sejong Cardiovascular Center between January 2010 and December 2018 if the three conditions of electrocardiography-proven atrial fibrillation, OAC therapy, and GI bleeding were met sequentially. Patients' medical history, demographics, laboratory findings, echocardiographic findings, and endoscopic treatment history were assessed. Information on anticoagulant type, doses, and timing; duration of cessation; and whether vitamin K injections were administered was collected. Individual thromboembolic and bleeding risks were calculated using the $CHA_2DS_2$-VASc, Glasgow-Blatchford, and HAS-BLED scoring systems [2,3]. The study protocol was received from the institutional review board of Sejong General Hospital (SGH 2018-08-027-001). Written informed consent was obtained from all patients, and we complied with the Declaration of Helsinki (6th revision).

### Definitions of cessation strategy

Anticoagulation cessation was decided individually in consultation with at least one GI specialist and one cardiologist. We simplified the cessation strategies into two groups. The following anticoagulation strategy was defined in the discontinuation group as follows. 1. For patients treated with vitamin K antagonist, 1) cessation for more than one day without a prothrombin

time (PT) international normalized ratio (INR) check; 2) cessation until INR <1.5 without bridging heparin therapy; and 3) cessation for more than seven days irrespective of heparin bridge therapy. 2. For patients treated with NOAC, 1) cessation for more than one day with normal kidney function; and 2) cessation for more than two days with impaired renal function [4,5]. If the cessation strategies did not meet the discontinuation group criteria described above, patients were classified as the continuation group.

Bridging heparin therapy was defined as the administration of unfractionated or low-molecular-weight heparin (LMWH) during anticoagulation withdrawal. The treatment target range for unfractionated heparin was 1.5- to 2.5-fold activated partial thromboplastin time prolongation. The full weight-adjusted dosing of LMWH was 0.5–1.0 anti-Xa units/mL [6].

We defined the high thromboembolic risk group as atrial fibrillation with a $CHA_2DS_2$-VASc score of >5, the presence of any mechanical cardiac valve, or a history of ischemic stroke.

### Primary outcomes

Primary thromboembolic outcomes were defined as a composite of ischemic stroke, systemic embolism including pulmonary embolism, deep vein thrombosis, myocardial infarction (ST-elevation myocardial infarction [STEMI] or non-STEMI with a troponin elevation >99th percentile of the upper reference limit), or any valvular thrombosis confirmed by echocardiography or computed tomography within 3 months of the index anticoagulation cessation [1,7]. The primary safety outcome was a composite of intracranial hemorrhage, acute bleeding resulting in a hemoglobin drop of >3 g/dL, or requiring the transfusion of more than two units of packed red blood cells (RBC) at 1 year.

### Data analysis and statistical methods

The characteristics of the two groups were compared using Pearson's chi-square or Fisher's exact tests for categorical variables and a two-sample t-test or Mann-Whitney U test for continuous variables. Categorical variables are expressed as numbers of objects and percentages. Continuous data are expressed as mean ± standard deviation. Statistical significance was set at P <0.05. Univariate stepwise logistic regression analysis was used to estimate the correlation between risk factors and each thromboembolic and bleeding event at 3 months and 1 year. However, a multivariate analysis could not be performed because the Glasgow-Blatchford and HAS-BLED scores included overlapping risk factors.

## Results

### Baseline characteristics by group

From January 2010 to December 2018, a total of 146 patients who met the enrollment criteria of this study were analyzed (Fig 1): 54 in the continuation group versus 92 in the discontinuation group. The patients' baseline characteristics are shown in Table 1. Patients in the discontinuation group were older (69.8 ± 9.0 vs. 74.9 ± 8.9 years, p = 0.001) and more likely diabetic (39.6% vs. 59.8%, p = 0.019). The mean $CHA_2DS_2$-VASc score was lower in the continuation group (3.4 ± 1.3 vs. 4.1 ± 1.6; p = 0.010). More patients in the continuation group underwent cardiac valve surgery (51.9% vs. 20.7%, p<0.001). The proportion of mechanical mitral valve replacements, either isolated or concomitant with aortic valve replacement, was significantly higher in the continuation group (p<0.001 and p = 0.022, respectively). Physicians more often chose to continue anticoagulation therapy among patients with mechanical mitral valves and GI bleeding (38.9% vs. 7.5%, p<0.001). The mean Glasgow-Blatchford and HAS-BLED scores

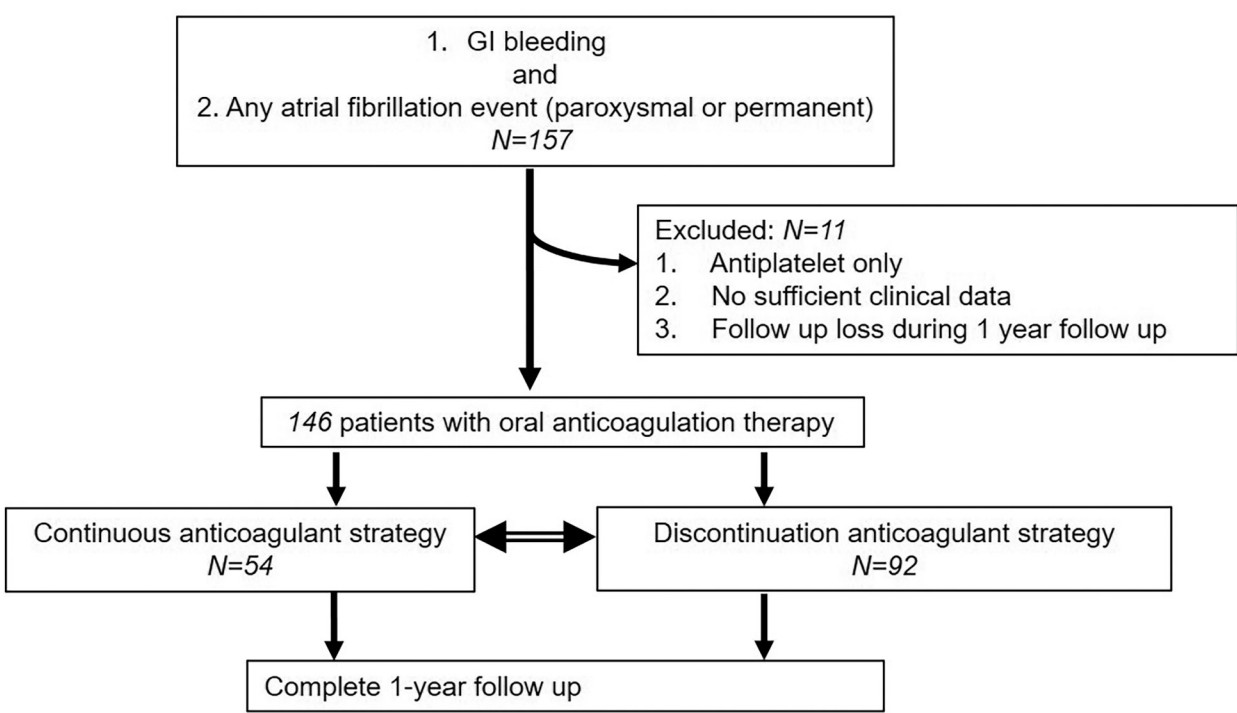

**Fig 1. Flowchart of patient enrollment.**

were lower in the continuation group (8.0 ± 2.4 vs. 8.9 ± 2.5, p = 0.037 and 2.3 ± 1.0 vs. 2.8 ± 1.0, p = 0.027, respectively).

Table 2 shows the characteristics, treatment strategies, and other clinical and endoscopic bleeding profiles of the patients with GI bleeding admitted to our hospital. There were no differences in vital signs at presentation. The vitamin K administration (33.3% vs. 40.2%, p = 0.407) rate was similar between groups, whereas the RBC transfusion (48.1% vs. 56.5%, p = 0.327) and fresh frozen plasma (FFP) transfusion (13.0% vs. 16.3%, p = 0.586) rates did not differ between them. However, more patients were treated by heparin bridge therapy (37.0% vs. 5.4%, p<0.001) and the mean OAC duration was shorter in the continuation group (2.2 ± 2.2 vs. 11.6 ± 21.4 days, p<0.001). When anticoagulant therapy was restarted, more patients in the continuation group used NOAC rather than vitamin K antagonists (14.8% vs. 41.3%; p<0.001). Esophagogastroduodenoscopy (EGD) was performed in 90.7% (49 of 54) of patients in the continuation group versus 89.1% (82 of 92) in the discontinuation group. Most of the bleeding cases were caused by gastric ulcer (36.7%), followed by gastritis (34.7%); in the continuation group, there was no significant difference in the discontinuation group (28.4% and 30.9%, respectively). Colonoscopy was performed in 31.4% (17 of 54) and 35.8% (33 of 92) of patients in each group. The most common source of bleeding during colonoscopy was colitis in both groups. Only a small number of patients in both groups (n = 2 [3.7%] vs. n = 1 [1.0%]) were treated with an invasive endoscopic procedure (cauterization, clipping, or epinephrine injection).

## Clinical outcomes

Fig 2 summarizes the patients with clinical events at 3 months and 1 year of follow-up. The bar indicator shows age, Glasgow-Blatchford score, and $CHA_2DS_2$-VASc score for each patient.

**Table 1. Clinical characteristics of patients with gastrointestinal bleeding according to the strategies for oral anticoagulation therapy.**

| | Continuation (n = 54) | Discontinuation (n = 92) | P-value |
|---|---|---|---|
| Age (years), mean ± SD | 69.8 ± 9.0 | 74.9 ± 8.9 | 0.001 |
| ≥ 75 years, n (%) | 17 (31.5) | 48 (52.2) | 0.015 |
| ≥ 65 years (%) | 41 (75.9) | 81 (88.0) | 0.056 |
| Female, n (%) | 34 (63.0) | 59 (64.1) | 0.887 |
| Current Smoking, n (%) | 3 (7.3) | 11 (17.2) | 0.147 |
| Hypertension, n (%) | 38 (71.7) | 73 (79.3) | 0.295 |
| DM, n (%) | 21 (39.6) | 55 (59.8) | 0.019 |
| Dyslipidemia, n (%) | 7 (13.5) | 23 (27.1) | 0.062 |
| Stroke, n (%) | 12 (22.6) | 28 (30.4) | 0.312 |
| CHF, n (%) | 24 (45.3) | 53 (57.6) | 0.152 |
| CKD, n (%) | 14 (26.4) | 32 (35.2) | 0.277 |
| eGFR, ml/min/1.73m$^2$ | 53.7 ± 29.3 | 51.1 ± 23.6 | 0.669 |
| Liver disease, n (%) | 6 (11.3) | 20 (22.0) | 0.109 |
| Valvular heart disease | 33 (61.1) | 29 (31.5) | <0.001 |
| Previous valve surgery, n (%) | 28 (51.9) | 19 (20.7) | <0.001 |
| Mitral repair, n (%) | 1 (1.9) | 3 (3.4) | >.99 |
| Mitral tissue valve, n (%) | 0 | 2 (2.2) | 0.531 |
| Aortic tissue valve, n (%) | 1 (1.9) | 0 | 0.370 |
| DVR with tissue valve, n (%) | 2 (3.7) | 4 (4.3) | >.99 |
| Mitral mechanical valve, n (%) | 15 (27.8) | 5 (5.4) | <0.001 |
| Aortic mechanical valve, n (%) | 3 (5.6) | 3 (3.3) | 0.670 |
| DVR with mechanical valve, n (%) | 6 (11.1) | 2 (2.2) | 0.022 |
| Any mechanical valve in MV, n (%) | 21 (38.9) | 7 (7.5) | <0.001 |
| PMV, n | 1 | 1 | |
| CHAD2DS2 VASc (mean ± SD) | 3.4 ± 1.3 | 4.1 ± 1.6 | 0.010 |
| Pacemaker insertion, n (%) | 9 (17) | 6 (6.8) | 0.058 |
| Previous Maze surgery, n (%) | 11 (20.8) | 11 (12.5) | 0.191 |
| PCI or CABG, n (%) | 6 (11.5) | 15 (17.4) | 0.364 |
| Antiplatelet use, n (%) | 13 (24.1) | 24 (26.1) | 0.787 |
| Statin use, n (%) | 18 (43.9) | 26 (35.6) | 0.383 |
| Echocardiographic EF (mean ± SD, %) | 49.6 ± 15.5 | 45.4 ± 13.3 | 0.544 |
| Admission duration, day | 7.8 ± 8.4 | 8.8 ± 11.3 | 0.618 |
| NOAC use, n (%) | 18 (33.3) | 25 (27.2) | 0.431 |
| Glasgow-Blatchford score (mean ± SD) | 8.0 ± 2.4 | 8.9 ± 2.5 | 0.037 |
| HAS-BLED score (mean ± SD) | 2.3 ± 1.0 | 2.8 ± 1.0 | 0.027 |

SD, standar deviation; DM, diabetes mellitus; CHF, congestive heart failure; CKD, chronic kidney disease; eGFR, estimated glomerular filtration rate; liver disease, cirrhosis or bilirubin > 2x normal value with aspartate aminotransferase/alanine aminotransferase/alkaline phosphatase >3x normal value; valvular heart disease, valvular stenosis or regurgitation > grade II/out of IV; DVR, dual valve replacement; MV, mitral valve; PMV, percutaneous mitral balloon valvuloplasty; PCI, percutaneous coronary intervention; CABG, coronary artery bypass surgery; EF, ejection fraction; NOAC, new oral anticoagulant.

Mid-cerebral arterial infarction occurred in two patients (#1 and #2), while bleeding events occurred in five patients (#2, #3, #4, #5, and #6). Patient #2 had two events simultaneously. He was a 75-year-old man with hypertension, diabetes, and previous stroke who currently smoked, was admitted with melena, and had a blood pressure of 90/60 mmHg, pulse rate of 82 bpm, hemoglobin 8.0 mg/dL, and PT INR = 2.0 at presentation. He had hemorrhagic gastritis (Forrest classification Ia) and was treated with four days of cessation of anticoagulation with bridge heparin therapy. The patient was discharged with a controlled PT INR = 2.42. However,

**Table 2. Characteristics of patients at presentation, treatment strategies and clinical and endoscopic bleeding profiles.**

| | Continuation (n = 54) | Discontinuation (n = 92) | P-value |
|---|---|---|---|
| Clinical presentation | | | 0.630 |
| Melena, n (%) | 20 (37.0) | 41 (44.6) | |
| Hematochezia, n (%) | 20 (37.0) | 32 (34.8) | |
| Anemia, n (%) | 5 (9.3) | 5 (5.4) | |
| Dyspnea, n (%) | 1 (1.9) | 5 (5.4) | |
| Hematemesis, n (%) | 3 (5.6) | 1 (1.1) | |
| Etc., n (%) | 5 (9.3) | 8 (8.7) | |
| Systolic BP at presentation, mmHg | 106.6 ± 16.1 | 109.1 ± 21.6 | 0.549 |
| Diastolic BP at presentation, mmHg | 65.0 ± 12.2 | 65.1 ± 12.2 | 0.974 |
| HR at presentation, /min | 80.6 ± 12.5 | 82.5 ± 20.8 | 0.589 |
| INR at presentation | 3.3 ± 1.4 | 3.5 ± 2.1 | 0.468 |
| Vitamin K use, n (%) | 18 (33.3) | 37 (40.2) | 0.407 |
| Heparin bridge therapy, n (%) | 20 (37.0) | 5 (5.4) | <0.001 |
| Oral anticoagulant cessation duration, day | 2.2 ± 2.2 | 11.6 ± 21.4 | <0.001 |
| Restart as NOAC, n (%) | 8 (14.8) | 38 (41.3) | 0.001 |
| FFP transfusion, n (%) | 7 (13.0) | 15 (16.3) | 0.586 |
| RBC transfusion, n (%) | 26 (48.1) | 52 (56.5) | 0.327 |
| EGD, n (%) | 49 (90.7) | 82 (89.1) | |
| Gastritis, n (%) | 17 (34.7) | 25 (30.9) | |
| Gastric ulcer, n (%) | 18 (36.7) | 23 (28.4) | |
| Forrest Ia/Ib, n | 1/0 | 1/3 | |
| Forrest IIa/IIb/IIc, n | 7/0/4 | 13/1/1 | |
| Forrest III, n | 6 | 7 | |
| Duodenal ulcer, n (%) | 4 (8.2) | 2 (2.4) | |
| Angioma, n (%) | 2 (4.1) | 4 (4.9) | |
| Malignant, n (%) | 1 (2.0) | 1 (1.2) | |
| No focus, n (%) | 6 (12.2) | 27 (33.3) | |
| Colonoscopy, n | 17 | 33 | 0.429 |
| Colitis, n (%) | 6 (35.2) | 11 (33.3) | |
| Angioma, n (%) | 0 | 1 (3.0) | |
| Small bowel origin or unknown, n (%) | 3 (17.6) | 3 (9.0) | |
| Malignancy, n (%) | 1 (5.8) | 2 (6.0) | |
| Polyp, n (%) | 1 (5.8) | 5 (15.1) | |
| hemorrhoid, n (%) | 4 (23.5) | 11 (33.3) | |
| ulcer, n (%) | 2 (11.7) | 0 | |
| Invasive intervention | | | |
| Cauterization/Clip/Epi injection, n | 1/0/1 | 0/1/0 | |

n, number; BP, blood pressure; HR, heart rate; INR, international normalized ratio of prothrombin time; NOAC, new oral anticoagulant; FFP, fresh frozen plasma; RBC, red blood cell; EGD, esophagogastroduodenoscopy; Epi, epinephrine.

44 days after the index admission, he suffered from a mid-cerebral arterial infarction and subsequent rebleeding, ultimately dying of ischemic colitis.

Table 3 shows the 3-month and 1-year clinical outcomes of the patients during follow-up. Thromboembolic events occurred in only two patients (3.7%) in the continuation group (p = 0.135). Recurrent bleeding events were comparable in two (3.7%) and three (3.3%) patients in each group (p>0.99). Mortality occurred in three (5.6%) and five (5.4%) patients, respectively, without a significant difference (p = 0.975).

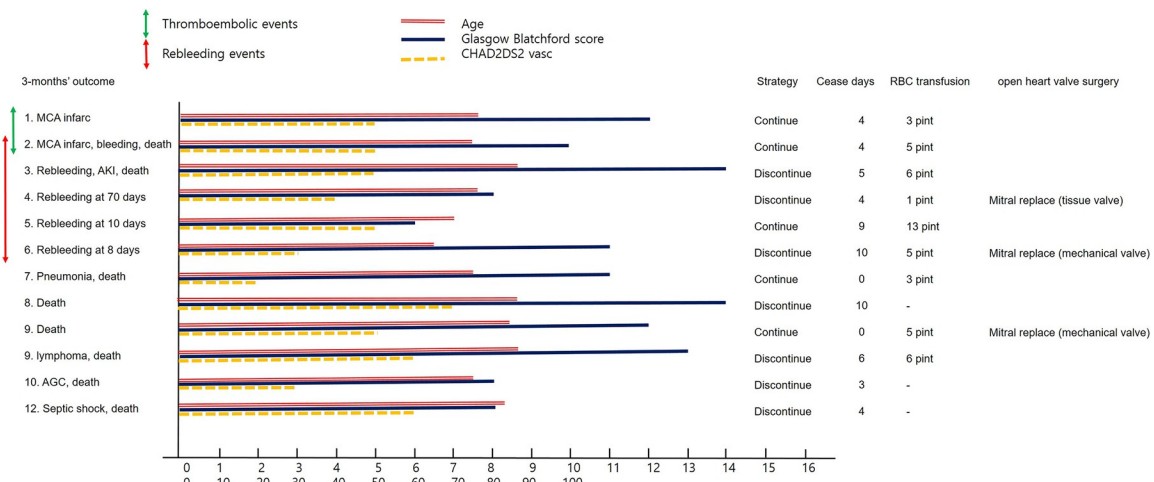

**Fig 2. Each individual's clinical thromboembolic and rebleeding events are drowned.** A green line with arrowheads at both ends indicates numbered patients who had thrombotic events (#1 and #2), while a red line with arrowheads indicates the patients had a rebleeding event (#2 to # 6). Age and Glasgow Blatchford score and the CHA$_2$DS$_2$-VASc score were drawn for each numbered patient. The information of the anticoagulation strategies, cessation duration, amount of RBC transfusion, and history of open-heart valve surgery are filled in this figure.

## Clinical outcomes of high thromboembolic risk patients

According to our definition, 33 of 54 patients (61.1%) in the continuation group versus 45 of 92 (48.9%) in the discontinuation group were classified as a high-risk subset. Table 4 shows treatment strategies of the patient group with high-risk thromboembolism and their outcome. The baseline rates of anticoagulant use were similar between the two groups (30.3% vs. 26.7%, p = 0.724). Although FFP or RBC transfusion rates were similar between groups, more patients in the continuation group used heparin bridge therapy (42.4% vs. 6.7%, p<0.001), while fewer received vitamin K injection therapy (27.3% vs. 42.4%, p = 0.174). The mean OAC cessation duration was shorter in the continuation group (2.4 ± 2.3 vs. 7.1 ± 7.0 days, p<0.001). All thromboembolic events (two cases) occurred in the high-risk subset during the 3-month follow-up. Similarly, all four thromboembolic events at the 1-year follow-up occurred in the high-risk subset. Most of the recurrent bleeding and mortality cases occurred in the high-risk subset, and there was no significant difference between the two strategies.

**Table 3. Three-months' and one years' clinical outcomes of patients during follow up.**

|  | Continuation n = 54 | Discontinuation n = 92 | P-value |
|---|---|---|---|
| 3 months' clinical events person,(%) | 5 (9.3) | 7 (7.6) | 0.892 |
| Thromboembolism n,(%) | 2 (3.7) | 0 | 0.135 |
| Recurrent bleeding n,(%) | 2 (3.7) | 3 (3.3) | >0.99 |
| Mortality n,(%) | 3 (5.6) | 5 (5.4) | 0.975 |
| 1 years' clinical events person,(%) | 5 (9.3) | 9 (9.8) | 0.917 |
| Thromboembolism n,(%) | 2 (3.7) | 2 (2.2) | 0.585 |
| Recurrent bleeding n,(%) | 4 (7.4) | 5 (5.4) | 0.632 |
| Mortality n,(%) | 3 (5.6) | 6 (6.5) | 0.815 |

person, patients' number who had clinical events; n, number of events.

**Table 4. Treatment strategies of patient group with high-risk thromboembolism and their outcome.**

| Treatments | Continuation (n = 33) | Discontinuation (n = 45) | P-value |
|---|---|---|---|
| Baseline NOAC treatment, n (%) | 10 (30.3) | 12 (26.7) | 0.724 |
| Vitamin K, n (%) | 9 (27.3) | 19 (42.2) | 0.174 |
| Heparin bridge therapy, n (%) | 14 (42.4) | 3 (6.7) | <0.001 |
| Restart as NOAC, n (%) C | 3 (9.1) | 18 (40.0) | 0.002 |
| FFP transfusion, n (%) | 4 (12.1) | 6 (13.3) | 0.874 |
| RBC transfusion, n (%) | 18 (54.5) | 26 (57.8) | 0.776 |
| Oral anticoagulant cessation duration, day | 2.4 ± 2.3 | 7.1 ± 7.0 | <0.001 |
| Clinical events | Continuation (n = 33) | Discontinuation (n = 45) | P-value |
| 3 months' clinical events person,(%) | 4 (12.1) | 5 (11.1) | 0.890 |
| Thromboembolism n,(%) | 2 (6.1) | 0 | 0.176 |
| Recurrent bleeding n,(%) | 2 (6.1) | 2 (4.4) | >0.99 |
| Mortality n,(%) | 2 (6.1) | 4 (8.9) | >0.99 |
| 1 years' clinical events person,(%) | 4 (12.1) | 7 (15.6) | 0.667 |
| Thromboembolism n,(%) | 2 (6.1) | 2 (4.4) | >0.99 |
| Recurrent bleeding n,(%) | 3 (9.1) | 3 (6.7) | 0.694 |
| Mortality n,(%) | 2 (6.1) | 4 (8.9) | >0.99 |

NOAC, new oral anticoagulant; FFP, fresh frozen plasma; RBC, red blood cell.

## Scoring systems and association with thromboembolism and rebleeding

Univariate analysis of age, renal function, Glasgow-Blatchford score, and HAS-BLED score did not predict 3-month rebleeding events. Similarly, a univariate analysis of age, $CHA_2DS_2$-VASc score, and discontinuation strategy failed to predict thromboembolic events at 3 months.

In contrast, Glasgow-Blatchford score was a predictor of 1-year rebleeding events (odds ratio, 1.36; 95% confidence interval, 0.68–2.20; p = 0.028). The $CHA_2DS_2$-VASc score showed a strong trend (odds ratio, 1.71; 95% confidence interval, 0.92–3.20; p = 0.089) about 1-year thromboembolic events (Table 5).

## Discussion

The key message of the current study was that short-term thromboembolic events were not directly related to the anticoagulation cessation strategy if the decision was made under cardiologist supervision. Although there were no obvious factors that could directly predict this short-term event after GI bleeding, we may be able to predict long-term thromboembolism or rebleeding using the pre-existing predictive scoring system.

GI bleeding is a relatively common complication in patients receiving anticoagulant therapy. The prevalence of GI bleeding in patients requiring warfarin is reportedly 2.5–10.1% [8,9]. In most cases, it is mild and can be treated on an outpatient basis by adjusting the oral anticoagulant dose or temporarily stopping anticoagulant therapy. Generally, the symptoms are not severe and can be treated in outpatient clinics by adjusting the oral anticoagulant dose or temporarily stopping the anticoagulant therapy. However, catastrophic events can sometimes develop in two forms. First, it could be in the form of fatal bleeding, hypovolemic shock, or organ ischemia. Second, systemic embolization, especially stroke, may occur due to discontinuation of anticoagulant therapy.

**Table 5. Univariate binary logistic regression analyses for predicting 3 months' rebleeding.**

| Variable for 3-months rebleeding | OR | 95% CI | P-value |
|---|---|---|---|
| Age | 1.02 | 0.92–1.13 | 0.634 |
| eGFR | 0.96 | 0.91–1.01 | 0.171 |
| Glasgow Blatchford score | 1.24 | 0.86–1.79 | 0.248 |
| HAS-BLED score | 1.34 | 0.61–2.94 | 0.462 |
| Discontinuation strategy | 0.02 | 0.14–5.16 | 0.872 |
| Discontinuation days | 0.94 | 0.67–1.31 | 0.729 |
| Variables for 3-months thrombosis | OR | 95% CI | P-value |
| Age | 1.02 | 0.87–1.20 | 0.742 |
| CHAD2DS2 VASc | 1.55 | 0.65–3.67 | 0.317 |
| Discontinuation strategy | 0.008 | 0–1567 | 0.435 |
| Variable for 1-year rebleeding | OR | 95% CI | P-value |
| Age | 1.06 | 0.98–1.14 | 0.145 |
| eGFR | 0.97 | 0.94–1.00 | 0.128 |
| Glasgow Blatchford score | 1.36 | 1.03–1.80 | 0.028 |
| HAS-BLED score | 1.22 | 0.68–2.20 | 0.491 |
| Discontinuation strategy | 0.71 | 0.19–2.66 | 0.618 |
| Variables for 1-year thrombosis | OR | 95% CI | P-value |
| Age | 1.02 | 0.91–1.15 | 0.621 |
| CHAD2DS2 VASc | 1.71 | 0.92–3.20 | 0.089 |
| Discontinuation strategy | 0.57 | 0.08–4.10 | 0.584 |

eGFR, estimated glomerular filtration rate.

## Endoscopy for anticoagulant patients

For the management of massive bleeding, endoscopy is used to provide information to identify ways to most effectively respond to the bleeding. Based on the bleeding lesion's information, it may be possible to determine whether complete suppression of anticoagulation is required. In contrast, if local treatment is available during endoscopy, and the course of this treatment is satisfactory, it may be possible to maintain anticoagulant treatment [10]. Indeed, EGD was performed in most patients in the study, and colonoscopy was used to identify bleeding lesions in approximately half the population. However, local endoscopic treatment was limited. The results of the endoscopy show that the bleeding was not localized but generalized; hence, it was difficult to identify a reliable single focus of bleeding. Despite the massive bleeding, the proportion of arterial hemorrhage origins that could indicate procedural treatment was negligible. Another possible reason is that the low endoscopic active treatment rate could result from the preference of gastroenterogist for more conservative treatments. It might be reasonable to assume that GI specialists prefer more conservative treatment to active injection treatment because injection therapy can provide a new bleeding focus instead of complete treatment in patients with coagulopathy. Therefore, as inferred from our data, stabilizing the vital signs to prevent anemia and determining the time to resume anticoagulation might be more critical than arranging the endoscopic examination schedule in critical situations.

## High-risk thromboembolism subset

Since endoscopic local treatment is limited, the most effective treatment is to induce systemic hemostasis using transfusions of vitamin K or FFP for warfarin and reversal agents for NOAC.

However, complete reversal can lead to permanent neurological sequelae followed by embolic cerebral infarction, the second possible cause of catastrophic events discussed at the outset. Therefore, concomitant heparin bridging therapy or shortening of the outside of the therapeutic range (<INR 1.5) may be required for patients with high thromboembolic risk. Traditionally, those with a high CHADS2 score (>5), metallic mitral valve, prosthetic valve with atrial fibrillation, recent thromboembolic events, or thrombophilia have been regarded as high-risk thromboembolic subsets [11]. Because all enrolled patients had atrial fibrillation, we set the high-risk groups as patients with mechanical mitral valve, a high $CHA_2DS_2$-VASc score (>5), and a previous history of stroke. All 3-month thromboembolic events occurred exclusively in the high-risk subset, and most of the rebleeding cases and deaths occurred in this subset. Although more active treatment or short-term anticoagulant cessation with heparin bridge therapy was performed in the continuation group, only a small decrease in short-term thromboembolic or rebleeding events was achieved.

Interestingly, patients in the discontinuation group were more likely to undergo a change from warfarin to NOACs after the index event. The prevalence of GI bleeding between warfarin and NOACs has differed in recent randomized clinical trials. In general, the use of NOAC was associated with an increased risk of GI bleeding compared to warfarin; the risk that is similar to warfarin was reported only in patients treated with 110 mg of dabigatran or apixaban. GI bleeding rates of edoxaban and warfarin were 1.23 vs. 0.82% (p<0.001) at a low dose; 1.23 vs. 1.51% (p = 0.03) at a high dose; rivaroxaban and warfarin were 3.2% and 2.2% (p<0.001). While apixaban and warfarin were 0.76% and 0.86% (p = 0.37); 110mg of dabigatran and warfarin (1.12 vs. 1.02%, p = 0.43) [12–16]. Nevertheless, as the referring physicians in this study showed, NOAC was more selected than warfarin for high-risk patients after the bleeding events. The result seems to be the effect that the physicians have established from the previous randomized clinical trial results that NOAC had less fatal bleeding than warfarin. It might be necessary to study the risk of further GI bleeding in patients who previously switched from warfarin to NOAC.

### Patients with valve replace surgery

It is not surprising that more patients in the continuation group had valvular heart disease or prior valve surgery. This tendency of treatment selection was not observed in patients who underwent tissue valve surgery (mitral repair, mitral tissue valve, and aortic tissue valve surgery: p>0.99, 0.531, and 0.370, respectively). Even among patients who underwent mechanical valve surgery, mitral valve patients showed a strong preference for continuation. However, this preference was not observed in patients who underwent aortic mechanical valve surgery. Because of the high flow velocity at the aortic valve versus the mitral valve, a better outcome of anticoagulation therapy cessation would be expected from the referring physicians [17].

Because of the recent increase in NOAC use, this prevalence of fatal bleeding is much lower than that observed with conventional warfarin therapy [18]. Nevertheless, NOAC use in some patients, especially those with mechanical valves, is unstable compared to warfarin. [19] Consequently, anticoagulant therapy in patients undergoing valvular surgery should be entirely dependent on warfarin. It would be dangerous to reverse the anticoagulant action until the bleeding ceases completely. Hence, patients need to receive bridging heparin therapy, and monitoring is necessary until they reach the therapeutic range of INR [20]. Kuramatsu et al. recently reported on anticoagulation management after acute intracerebral hemorrhage (ICH) in patients requiring long-term OAC due to the presence of mechanical heart valves [6]. They recommended 6 days after ICH as the optimal timing of restarting anticoagulation therapy due to the lowest risk for thromboembolic and hemorrhagic complications. However,

physicians seem to apply more flexible therapeutic indications for GI bleeding patients with mechanical valves, such as early initiation of OAC therapy or adequate heparin bridge therapy. In our study, discontinuation days were not associated with short-term rebleeding rates. With the case of previous ICH and GI bleeding, physicians' treatment strategies should differ. Further studies on anticoagulant treatment strategies are needed for further clarification.

## Risk factors for thromboembolism or rebleeding during anticoagulation

There were no definite factors associated with rebleeding or thromboembolism at 3 months on univariate analysis. In contrast, known risk stratification systems are closely related to 1-year outcomes. It may be predicted that the longer the observation period, the higher the relevance to this prediction system. The high $CHA_2DS_2$-VASc score showed a strong trend toward 1-year thromboembolic events, while the Glasgow-Blatchford score was a predictor of 1-year rebleeding events. It would be reasonable to use these scoring systems as future risk assessment indicators for high-risk patients with GI bleeding.

## Limitation

Our study had several limitations. First, its sample size was small. Second, group comparisons were not conducted using randomized data. However, we believe that systematic classification of anticoagulant strategy, a high EGD performance rate, and detailed individual event records will provide sufficient information to overcome these weaknesses.

## Conclusion

Whether to maintain anticoagulants based on cardiologist and GI specialist assessment did not affect the occurrence of adverse events in the cardiac patients after GI bleeding. The risk of further thrombotic or rebleeding should be controlled individually based on a pre-existing stratification system.

## Supporting information

**S1 Data.**
(XLSX)

## Author Contributions

**Conceptualization:** Dongyoung Lee, Tae-Hoon Kim.

**Formal analysis:** Tae-Hoon Kim.

**Investigation:** Je Sang Kim, Hyun-Jong Lee, Ji Bak Kim, Ji-young Kim.

**Resources:** Ho-Jun Jang, Dongyoung Lee, Tae-Hoon Kim.

**Writing – original draft:** Ho-Jun Jang, Dongyoung Lee.

**Writing – review & editing:** Ho-Jun Jang, Tae-Hoon Kim.

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
