## [Decision Letter · Decision Letter 0]

18 Apr 2022

PONE-D-22-07554Clinical outcomes of gastrointestinal bleeding management during anticoagulation therapyPLOS ONE

Dear Dr. Kim,

Thank you for submitting your manuscript to PLOS ONE. After careful consideration, we feel that it has merit but does not fully meet PLOS ONE’s publication criteria as it currently stands. Therefore, we invite you to submit a revised version of the manuscript that addresses the points raised during the review process.

We look forward to receiving your revised manuscript.

Kind regards,

Tariq Jamal Siddiqi

Academic Editor

PLOS ONE

Journal Requirements:

 [The funders had no role in study design, data collection and analysis, decision to publish, or preparation of the manuscript.]

Reviewers' comments:

Reviewer's Responses to Questions

**Comments to the Author**

1. Is the manuscript technically sound, and do the data support the conclusions?

Reviewer #1: Yes

2. Has the statistical analysis been performed appropriately and rigorously? 

Reviewer #1: Yes

3. Have the authors made all data underlying the findings in their manuscript fully available?

Reviewer #1: Yes

4. Is the manuscript presented in an intelligible fashion and written in standard English?

Reviewer #1: Yes

5. Review Comments to the Author

Reviewer #1: 1. Elaborate upon the Introduction and Discussion sections, discussing the rationale and the previous literature in greater depth. Keep the Discussion concise as well by ensuring that material in the Results section of the study are not frequently repeated in the Discussion section.

2. Ensure that units for all parameters are mentioned. For example, units of eGFR and admission duration should be mentioned in the tables.

3. Discuss any bias that may have arisen and highlight future areas in which potential work/research may be done.

6. PLOS authors have the option to publish the peer review history of their article (what does this mean?). If published, this will include your full peer review and any attached files.

Reviewer #1: **Yes: **Warda Ahmed

---

## [Author Response · Author response to Decision Letter 0]

15 May 2022

Reviewer #1: 

1. Elaborate upon the Introduction and Discussion sections, discussing the rationale and the previous literature in greater depth. Keep the Discussion concise as well by ensuring that material in the Results section of the study are not frequently repeated in the Discussion section.

Response) Thank you for your review. We sincerely appreciate your high insight into our study. We changed and rearranged the introduction and discussion sections according to your recommendation. We made this section more concise and eliminated the repetition of results.

2. Ensure that units for all parameters are mentioned. For example, units of eGFR and admission duration should be mentioned in the tables.

Response) Thank you very much for the detailed point. As you pointed out, we added units for all parameters in the table

3. Discuss any bias that may have arisen and highlight future areas in which potential work/research may be done.

Response) In the Discussion, we added the following sentence at the High-risk thromboembolism subset section (line 280~284). “The result seems to be the effect that the physicians have established from the previous randomized clinical trial results that NOAC had less fatal bleeding than warfarin. It might be necessary to study the risk of further GI bleeding in patients who previously switched from warfarin to NOAC.” We added the following sentence in the Patients with valve replace surgery section (line 307~309). “With the case of previous ICH and GI bleeding, physicians’ treatment strategies should differ. Further studies on anticoagulant treatment strategies are needed for further clarification.”

4. We rearranged the discussion section; thus, the orders of references were changed. We add two references (#10 and #20).

Response) “10. Barkun AN, Almadi M, Kuipers EJ, Laine L, Sung J, Tse F, et al. Management of Nonvariceal Upper Gastrointestinal Bleeding: Guideline Recommendations From the International Consensus Group. Ann Intern Med. 2019;171(11):805-22. Epub 2019/10/22. doi: 10.7326/M19-1795. PubMed PMID: 31634917; PubMed Central PMCID: PMCPMC7233308.

20. Delate T, Meisinger SM, Witt DM, Jenkins D, Douketis JD, Clark NP. Bridge Therapy Outcomes in Patients With Mechanical Heart Valves. Clin Appl Thromb Hemost. 2017;23(8):1036-41. Epub 2016/09/23. doi: 10.1177/1076029616669786. PubMed PMID: 27655997.”

---

## [Editor Report · Decision Letter 1]

18 May 2022

Clinical outcomes of gastrointestinal bleeding management during anticoagulation therapy

PONE-D-22-07554R1

Dear Dr. Kim,

We’re pleased to inform you that your manuscript has been judged scientifically suitable for publication and will be formally accepted for publication once it meets all outstanding technical requirements.

Kind regards,

Tariq Jamal Siddiqi

Academic Editor

PLOS ONE
---

## [Editor Report · Acceptance letter]

27 May 2022

PONE-D-22-07554R1 

Clinical outcomes of gastrointestinal bleeding management during anticoagulation therapy 

Dear Dr. Kim:

I'm pleased to inform you that your manuscript has been deemed suitable for publication in PLOS ONE. Congratulations! Your manuscript is now with our production department. 

Kind regards, 

on behalf of

Dr. Tariq Jamal Siddiqi 

Academic Editor

PLOS ONE